# VATS versus Open Lobectomy following Induction Therapy for Stage III NSCLC: A Propensity Score-Matched Analysis

**DOI:** 10.3390/cancers15020414

**Published:** 2023-01-08

**Authors:** Kheira Hireche, Youcef Lounes, Christophe Bacri, Laurence Solovei, Charles Marty-Ané, Ludovic Canaud, Pierre Alric

**Affiliations:** 1Department of Thoracic and Vascular Surgery, Arnaud de Villeneuve University Hospital, 34090 Montpellier, France; 2PhyMedExp, INSERM, CNRS, University of Montpellier, 34295 Montpellier, France

**Keywords:** lung cancer, stage III, lobectomy, VATS

## Abstract

**Simple Summary:**

Although video-assisted thoracoscopy surgery is now considered the standard treatment for early-stage lung cancer, the relevance of VATS in locally advanced lung cancer remains unknown. Several studies have been conducted to assess the feasibility and safety of VATS lobectomy for locally advanced NSCLC. However, only a handful have used propensity score matching to compare the operative and oncologic outcomes of VATS versus open lobectomy. Furthermore, these studies included a mixture of stages (II, III, and IV) and did not particularly evaluate the significance of VATS in the treatment of stage III disease. In this study, we compared the perioperative and oncologic outcomes of VATS with open lobectomy for stage III NSCLC and used propensity score matching to produce a well-balanced cohort of patients undergoing VATS and open lobectomy in order to minimize selection bias and achieve convincing statistical results.

**Abstract:**

Objectives: This study aims to evaluate the perioperative and oncologic outcomes of thoracoscopic lobectomy for advanced stage III NSCLC. Methods: We retrospectively reviewed 205 consecutive patients who underwent VATS or open lobectomy for clinical stage III lung cancer between January 2013 and December 2020. The perioperative and oncologic outcomes of the two approaches were compared. Long-term survival was assessed using the Kaplan–Meier estimator. Propensity score-matched (PSM) comparisons were used to obtain a well-balanced cohort of patients undergoing VATS and open lobectomy. Results: VATS lobectomy was performed in 77 (37.6%) patients and open lobectomy in 128 (62.4%) patients. Twelve patients (15.6%) converted from VATS to the open approach. PSM resulted in 64 cases in each group, which were well matched according to twelve potential prognostic factors, including tumor size, histology, and pTNM stage. Between the VATS and the open group, there were no significant differences in unmatched and matched analyses, respectively, of the overall postoperative complications (*p* = 0.138 vs. *p* = 0.109), chest tube duration (*p* = 0.311 vs. *p* = 0.106), or 30-day mortality (*p* = 1 vs. *p* = 1). However, VATS was associated with shorter hospital stays (*p* < 0.0001). The five-year overall survival (OS) and five-year Recurrence-free survival (RFS) were comparable between the VATS and the open groups. There was no significant difference in the recurrence pattern between the two groups in both the unmatched and matched analyses. Conclusion: For the advanced stage III NSCLC, VATS lobectomy achieved equivalent postoperative and oncologic outcomes when compared with open lobectomy without increasing the risk of procedure-related locoregional recurrence.

## 1. Introduction

Non-small cell lung cancer (NSCLC) is the leading cause of cancer-related death worldwide. Unfortunately, the advanced-stage (stages III and IV) disease accounts for 79% of newly diagnosed patients, and the five-year survival rate ranges from 4 to 28% [1], depending on the disease stage, the patient’s medical conditions, and treatment modality. Chemoradiotherapy has traditionally been the mainstay of treatment for advanced NSCLC, while current guidelines recommend therapeutic intent pulmonary resection for patients with resectable stage IIIA disease and oligometastatic stage IV disease [2]. However, recent studies point to the benefits of surgical resection for more advanced-stage cases (which have typically been considered unresectable [3,4,5,6]) and have indicated that such intervention can offer significant improvements for both hospital mortality and long-term survival.

The benefits of video-assisted thoracoscopic (VATS) lobectomy in terms of reduced morbidity, faster resumption of daily activities [7], and even higher OS and RFS [8] are well established in patients with early-stage NSCLC. As a result, VATS is now considered to be the standard treatment for early-stage NSCLC [2]. Despite better compliance with adjuvant chemotherapy following a VATS procedure in patients with advanced disease [9] and the expansion of the use of thoracoscopic procedures for more technically challenging operations, the benefits of VATS for patients with advanced-stage NSCLC have yet to be clearly defined. Indeed, several studies have reported the use of VATS lobectomy for locally advanced NSCLC [10,11,12,13,14] and indicate that VATS is both feasible and safe and is associated with a reduced hospital stay and chest tube duration with equal oncologic efficacy as compared to open lobectomy. However, these studies involved a mixture of disease stages, which may have biased the results in favor of the VATS approach, nor did these studies specifically assess the role of VATS in the management of stage III disease. Consequently, the present study was designed to compare the perioperative and oncologic outcomes of VATS and open lobectomy for patients with advanced stage III NSCLC. We performed propensity score matching to create two homogenous groups for comparison, minimize selection bias, and achieve convincing statistical results.

## 2. Material and Methods

### 2.1. Patient Selection

We retrospectively analyzed 205 consecutive patients with advanced stage III NSCLC who underwent a lobectomy between January 2013 and December 2020 at the Department of Thoracic and Vascular Surgery, Arnaud de Villeneuve Teaching Hospital, Montpellier. The 8th edition of the Union for International Cancer Control’s tumor node metastasis (TNM) staging system was used to determine clinical and pathological stages. The preoperative staging was assessed by a computed tomographic (CT) scan of the chest, abdomen, and pelvis, as well as a positron emission tomographic (PET-CT) scan, brain imaging with a CT scan, or magnetic resonance imaging (MRI), and bronchoscopy. Mediastinoscopy or endobronchial ultrasonography (EBUS) was used for mediastinal staging. Patients with stage I–II NSCLC and those with tumors other than NSCLC were excluded. Patients with central tumors requiring complex bronchovascular reconstruction were offered an open approach. Peripheral tumors less than 7 cm were managed by VATS. Since 2015, in the case of chest wall invasion, a hybrid approach was performed as previously described [15]. All patients received induction therapy. The preoperative choice of neoadjuvant-type therapy was based on the patient’s conditions, their physician’s recommendations, and the availability of induction therapy protocols. A chest CT and/or PET-CT scan were used to evaluate the response. In patients with cN2 (single or bulky) and cN-3 disease, invasive restaging using mediastinoscopy or EBUS was performed after induction therapy and prior to resection. Patients with downstaging and without progression underwent radical resection. Four surgeons with at least 4 years of VATS experience were involved in the study. VATS lobectomy was performed through three ports without rib spreading. Open lobectomy was performed via a posterolateral thoracotomy through the fifth interspace with latissimus dorsi muscle section and rib spreading. Patients were divided into two groups according to the surgical approach adopted: a VATS group and an open group. Systemic lymph node dissection of all hilar (N1) and at least three mediastinal (N2) nodal stations were routinely performed. In postoperative care, all patients were managed according to a standardized postoperative protocol. Chest drains were typically removed when there was no air leakage and their volume was less than 400 cc/day. This study was approved by the Institutional Review Board, with individual patient consent being waived.

### 2.2. Data Extraction

Data were collected on patient demographics, smoking history, comorbidities, pulmonary function test, clinical stage, tumor location and size as measured by CT, neoadjuvant therapy modality, histological type, pathological stage, number of lymph nodes and stations removed, surgical details, chest tube duration, length of hospitalization, reoperation, and postoperative complications included any of the following: pneumonia, atrial fibrillation, prolonged air leak (more than 5 days postoperatively), adult respiratory distress syndrome (ARDS), bronchopleural fistula, pleural effusion, and heart failure. Perioperative mortality was defined as death within 30 days of the operation.

Follow-up data were collected from clinical notes and direct contact with patients and physicians. Overall survival (OS) was defined as the time period between surgery and death from any cause or the last follow-up evaluation. Recurrence-free survival (RFS) was defined as the period between surgery and recurrence or death from any cause.

### 2.3. Statistical Analysis

Statistical analysis was performed using SPSS IBM software version 25 (IBM, Armonk, NY, USA). The baseline characteristics and outcomes of the VATS and open groups were compared using Pearson’s χ2-test or Fisher’s exact test when applicable for categorical variables and Student’s unpaired *t*-test or the Wilcoxon rank-sum test when applicable for continuous variables. Patients who underwent conversions from VATS to open lobectomy were assessed using an intent-to-treat analysis and, for the purposes of our analysis, continued to be attributed to the VATS group. OS and RFS were evaluated using the Kaplan–Meier method and the log-rank test. Propensity score matching (PSM) was performed to balance the confounding factors between the two groups to minimize potential selection bias. Propensity scores were developed and defined as the probability of treatment with the VATS approach versus the open approach conditional on measured covariates. Variables included in the propensity score model were: age, sex, hypertension, coronary artery disease, diabetes, COPD, smoking history, FEV1, clinical tumor size, pathological T and N status, tumor location, histology, and induction chemoradiotherapy. Patients were then matched on the propensity score using a 1:1 nearest neighbor matching algorithm with a caliper distance of 0.01 and no replacement. Following propensity matching, the patient demographics and outcomes were assessed using Pearson’s χ2-test or Fisher’s exact test when applicable for categorical variables and Student’s unpaired *t*-test or the Wilcoxon rank-sum test when applicable for continuous variables. OS and RFS were evaluated using the Kaplan–Meier method and the log-rank test. Furthermore, the Cox proportional hazards model was used to identify the independent prognostic factors of RFS for these patients. All tests were two-sided, using an alpha of < 0.05 to be considered statistically significant.

## 3. Results

### 3.1. Unmatched Population

#### 3.1.1. Patient Characteristics

The clinical and pathologic backgrounds are described in Table 1 and Table 2, respectively.

A total of 205 patients were enrolled in the study, of whom seventy-seven had undergone VATS lobectomy and one-hundred twenty-eight underwent open lobectomy. Compared with the open group, patients in the VATS group were older (61.33 ± 8.54 vs. 57.6 ± 9.52, *p* = 0.031) and less likely to have diabetes (9.1 vs. 13.3%, *p* = 0.041) and cardiac disease (6.5 vs. 16.4%, *p* = 0.039). Preoperative forced expiratory volume in one second (FAV1%) was significantly lower in the VATS group (88.94 ± 16.98 vs. 90.50 ± 16.64%, *p* = 0.008), but diffusion capacity of the lung for carbon monoxide (DLCO%) was similar for both groups. The mean clinical tumor size was smaller in the VATS group than in the open group (31.16 ± 17.33 vs. 54.12 ± 35.77 mm, *p* = 0.002). There were no significant differences between the two groups in terms of histology, induction therapy, and anatomic distribution of resected lobes. However, patients in the open group had a larger pathological tumor size (42.3 ± 30 vs. 31.4 ± 18.9 mm, *p* = 0.002) and a higher pathological T (*p* < 0.001) and N (*p* = 0.013) status. The number of lymph node stations harvested and the total number of removed lymph nodes were similar for the two procedures. However, the VATS approach achieved a greater resection R0 than the open approach (94.8 vs. 82%, *p* = 0.012) (Table 2).

#### 3.1.2. Perioperative Outcomes

Perioperative outcomes for the two groups are listed in Table 3. Twelve patients (15.6%) were converted from VATS to open surgery. The reasons for conversions were fibrotic tissue and tight adhesions (six patients), hilar anthracofibrotic nodes (two patients), and extensive pulmonary artery involvement requiring bypass reconstruction (four patients). None of the conversions required more extensive resection than expected, and none led to perioperative death. There were no significant differences in surgery time, intra-operative blood loss, 30-day mortality, or overall morbidity, including bronchopleural fistula, prolonged air leak, atrial fibrillation, postoperative pneumonia, and respiratory failure. There was no difference in postoperative chest tube drainage duration between the two groups, but the length of hospital stay was significantly shorter (4 vs. 8 days, *p* < 0.0001) in the VATS group. 

### 3.2. Oncologic Outcomes

The median follow-up period was 23.5 months for the VATS group (range 1–69 months) and 26.3 months (range 1–82 months) for the open group. The five-year OS and RFS was 55.7% (95% CI, 42.3–59.5) and 42.8% (95% CI, 31–53%) in the VATS group, and 51.8% (95% CI, 47.2–59.6) and 31% (95% CI, 29–46%) in the open group, respectively. There was no significant difference in the five-year OS (log-rank test, *p* = 0.563) or five-year RFS (log-rank test, *p* = 0.193) between the two groups (Figure 1). In addition, no significant differences in overall, local, regional or distant recurrence were found (*p* = 0.356, see Table 4).

### 3.3. Matched Population

#### 3.3.1. Patient Characteristics and Perioperative Outcomes

The PSM created 64 cases of VATS lobectomy and 64 cases of open lobectomy. The baseline characteristics of the matched patients are listed in Table 1. Both groups were similar in age, sex, smoking history, FEV1, DLCO, comorbidities, clinical tumor size, pathologic stage, histology, anatomic distribution of resected lobe, and induction chemoradiotherapy. After propensity matching, the incidence of overall postoperative complications was similar (*p* = 0.109). VATS lobectomy was also associated with less blood loss (140 mL vs. 195 mL, *p* = 0.023) and shorter length of hospital stay (4 vs. 7 days, *p* < 0.0001). However, there remained no significant difference in surgery time, chest tube drainage duration, and 30-day mortality between the two groups.

After matching, more lymph node stations were harvested in the VATS group than the open group (5.9 ± 1.7 vs. 4.8 ± 1.6, *p* = 0.011), but the total number of removed lymph nodes was almost similar between the two procedures (12.5 ± 5.3 vs. 11.5 ± 6.3, *p* = 0.219). There was no difference between the two groups in terms of resection margins R0 (93.7 vs. 90.6, *p* = 0.443, Table 2).

#### 3.3.2. Oncologic Outcomes

The median follow-up of the matched population was 21.4 months for the VATS group and 23.7 months for the open group. The Kaplan–Meier analysis revealed no significant differences in the five-year OS [62.6% (CI 95%, 49.6–71.4) vs. 52.2 % (CI 95%, 47.8–73.5), log-rank test *p* = 0.622] and five-year RFS [44.9% (CI 95%, 31.2–50.3) vs. 32.5% (CI 26.2–50), log-rank test *p* = 0.355] between the VATS and the open groups, respectively (Figure 2). The pattern of recurrence was similar between the two groups (Table 4). 

The multivariable-adjusted survival analysis revealed that sex (HR, 0.349; 95% CI, 0.134–0.911; *p* = 0.031) and pathologic TNM stage (HR, 1.092; 95% CI, 1.050–1.136; *p* < 0.001) were independent predictors of worse RFS, but the VATS approach was not (HR, 2.359; 95% CI, 0.946–2.883; *p* = 0.066; Table 5).

## 4. Discussion

In this comparative study, we evaluated the perioperative and oncologic outcomes of VATS lobectomy for stage III NSCLC and found that the VATS approach was associated with shorter hospital stays compared with open lobectomy, whereas no significant difference was identified in perioperative outcomes, including perioperative complications and 30-day mortality. Additionally, there was no significant difference in five-year OS and RFS between the groups in both unmatched and matched populations. These results support the non-inferiority of VATS over open lobectomy.

Until recently, mini-invasive surgery was considered a contraindication for advanced-stage NSCLC resection, owing to concerns regarding its safety, the technical challenges of hilar dissection after induction therapy [16], and uncertainties related to the completeness of oncologic resection. However, with developments in thoracoscopic equipment and growing clinical experience, some thoracic surgeons have expanded the indications of VATS to advanced stages of lung cancer, thus providing evidence of its safety and efficiency [14,17,18,19]. Despite the obvious contribution of these studies to the mainstreaming of VATS, they mostly include only stage II or IIIA patients without applying a propensity analysis. To date, few studies have compared short- and long-term outcomes of VATS versus open lobectomy in a well-balanced population of patients with advanced-stage NSCLC. Cao et al. [20] conducted a large multi-institutional study to construct a propensity score analysis for VATS versus open lobectomy and matched a total of 2916 patients, of whom six-hundred sixty-five had stage IIIA NSCLC. While this analysis concluded that VATS has similar long-term survival outcomes compared to open lobectomy, it failed to compare perioperative and recurrence-free survival data for the two groups. Another retrospective study by Yang et al. [13] reviewed 272 patients who had undergone neoadjuvant therapy and matched a total of forty-five patients with stage III–IV NSCLC. The authors reported a 10% conversion rate and no significant differences between the two groups in 30-day mortality, overall morbidity, and 3-year OS and RFS. More recently, Chen et al. [12] reported the results of 120 pairs of well-matched VATS and open lobectomy in advanced-stage NSCLC patients, of whom approximately one-third only had stage IIIA disease. This study showed that VATS lobectomy was associated with a significantly shorter hospital stay and better compliance for adjuvant chemotherapy but no significant decrease in blood loss, chest tube drainage duration, or postoperative complications. The present study’s findings are largely aligned with those of Chen et al. [12].

The 15.6% conversion rate was relatively high in our series, which may be attributed to the neoadjuvant therapy. In a large, multi-institutional, propensity-matched study of 2887 patients who underwent either VATS or open lobectomy after neoadjuvant therapy, Yang et al. [21] reported a 20% conversion rate following induction chemotherapy and a 25% rate following induction chemoradiation. In line with our findings, there was no significant difference in perioperative mortality between patients who underwent conversions or open lobectomy. These results may eliminate persistent apprehension about the safety of VATS for advanced-stage NSCLC. Nevertheless, it must be acknowledged that the avoidance of intra-operative life-threatening injuries during the VATS procedure for advanced disease, especially following induction therapy, requires a careful selection of patients and proactive conversions to thoracotomy where doubt exists to probable injury. The criteria by which thoracotomy and VATS candidates might be distinguished has yet to be clarified because of the wide spectrum of patients generally involved and also because of the within-stage heterogeneity. We believe that the VATS approach should be reserved for tumors ≤ 7 cm and without massive hilar invasion. Chest wall involvement or N2 disease should not hamper the VATS procedure, even after induction therapy [15,22]. 

In our study, while propensity score matching showed significantly decreased bleeding volume in the VATS group, no differences were identified between the two groups in chest tube duration, overall complications, or early mortality (i.e., within 30 days). This study’s most interesting finding was the reduced length of hospital stay for the VATS group, both in matched and unmatched analyses, despite the two groups showing a similar overall rate of postoperative complications and chest tube duration. One possible explanation is the reduced trauma leading to less postoperative pain and lighter psychological burden in the VATS group allowing faster recovery than open lobectomy. While previous studies have highlighted the advantages of this [7,23], we believe that a VATS approach should be particularly emphasized for frail patients with an advanced stage of the disease who require multi-modality therapy. 

The reliability and accuracy of the VATS procedure in achieving adequate lymph node dissection have long been a matter of controversy. All surgeons are aware of the complexity and the prognostic impact of achieving a thorough lymph clearance, especially in patients with advanced-stage disease or lymph node metastasis. Several studies have evaluated the role of VATS lobectomy for stage I NSCLC cN0-pN2 and have reported a similar rate of lymph node upstaging and five-year recurrence-free survival for both approaches [24,25,26,27], which points to the oncologic efficacy of VATS lymphadenectomy. The present study found that the number of lymph node stations harvested by VATS was superior to thoracotomy in the matched population (*p* = 0.011) without impacting the total number of lymph nodes removed (*p* = 0.219). Consistent with other studies [28,29], these results may be attributed to the magnification of the surgical field by camera resulting in a clearer visualization of the anatomical structures and lymph node stations. The growing skills and experience accumulated in our institution, thanks to years of VATS practice, could also be a factor [30]. The earlier works of D’Amico et al. [31] and Lee et al. [32] show that cumulative experience may positively affect the accuracy of the nodal resection. 

Complete resection is a major prognostic factor and may increase the survival of advanced-stage lung cancer [33,34]. In our study, the negative margin rate of the VATS procedure was significantly higher than that of open lobectomy in the entire cohort, but not after matching. This latter fact is probably related to the smaller tumor size in the VATS group rather than to the procedure itself. Furthermore, in both unmatched and propensity score-matched cohorts, we found no significant differences in five-year OS and RFS between the VATS and open groups. More interesting still, the two groups’ recurrence pattern was similar even after propensity matching. A limited number of studies have addressed the seldom-debated question of recurrence patterns following VATS lobectomy [35,36,37]. These studies involved patients with early-stage disease, and all reported a similar recurrence rate between VATS and open lobectomy and concluded that the VATS procedure does not increase the risk of procedure-related locoregional recurrence. Our results are strikingly consistent with the above studies, despite our cohorts with more advanced-stage disease. As such, our results may help to dispel lingering doubts regarding the ability of VATS to achieve oncologic outcomes equivalent to those of standard thoracotomy. 

## 5. Limitations

This study had several limitations. First, this was a single-center retrospective cohort study rather than a randomized controlled trial; this could be a source of unobserved confounding selection bias between the two groups. While we used the PSM to balance the observable variables between the two groups, a number of potential unknown factors could not be adjusted, thus reducing the verification effectiveness of our study. Second, it was not easy to eliminate selection bias since only patients who showed a favorable response and good tolerance to initial treatments were, by definition, sufficiently and medically fit to then undergo surgery. Third, there is an obvious surgeon choice bias privileging an open approach to larger, complicated central tumors. We tried to address this by matching tumor size and location, and given the evident inability to carry out a large-scale randomized study, we believe this propensity analysis minimized bias as much as possible. Fourth, the criteria used for assessing the efficacy of neoadjuvant therapy were not fully consistent; for example, invasive mediastinal restaging was only performed for patients with cN2-N3 prior to induction therapy, which might undermine the oncologic outcomes. Next, given this study’s small sample size, we were unable to perform further stratified analysis, which would consider neoadjuvant regimens. Lastly, the follow-up period was relatively short, which limits long-term conclusions from being drawn. 

## 6. Conclusions

The current study has demonstrated that, in experienced VATS centers, VATS lobectomy for the treatment of advanced stage III NSCLC is safe, reliable, and associated with a shorter hospital stay and equivalent oncological outcomes compared with standard thoracotomy. We believe that in selected patients without huge and/or central tumors, VATS may be a sound alternative to thoracotomy, especially for patients weakened by muti-modality treatment. The patient selection process is also fundamental to avoid untimely conversions

## Figures and Tables

**Figure 1 cancers-15-00414-f001:**
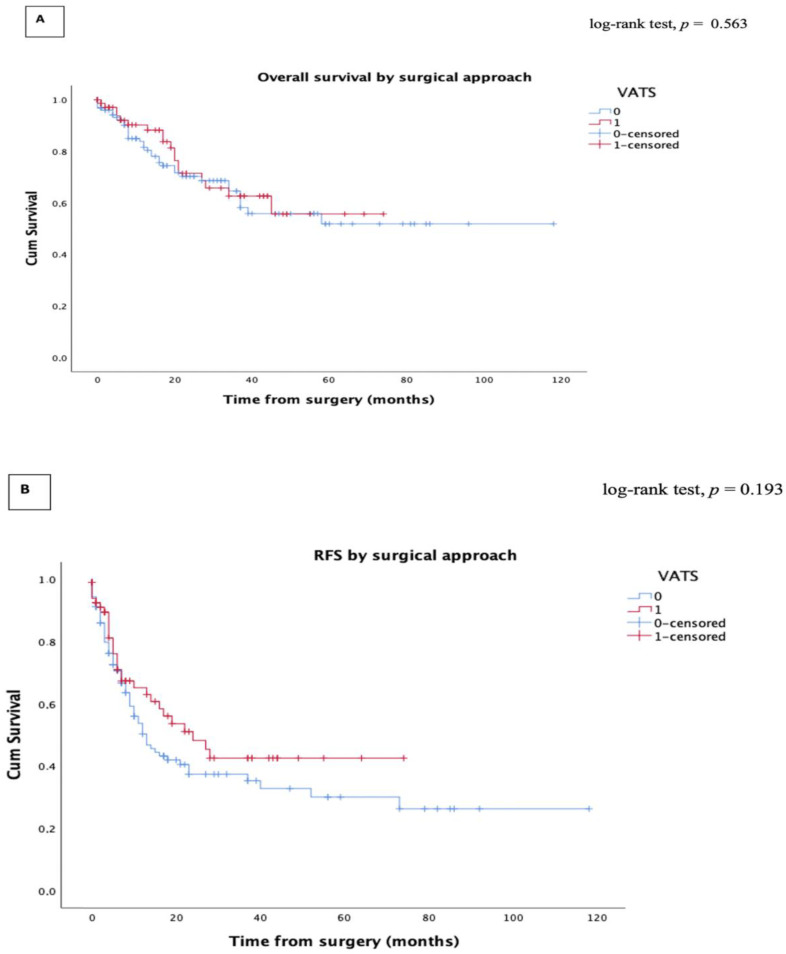
Kaplan–Meier curve for the long-term survival of the entire cohort. (**A**) Overall survival; (**B**) Recurrence-free survival. VATS: Video-assisted thoracoscopic surgery.

**Figure 2 cancers-15-00414-f002:**
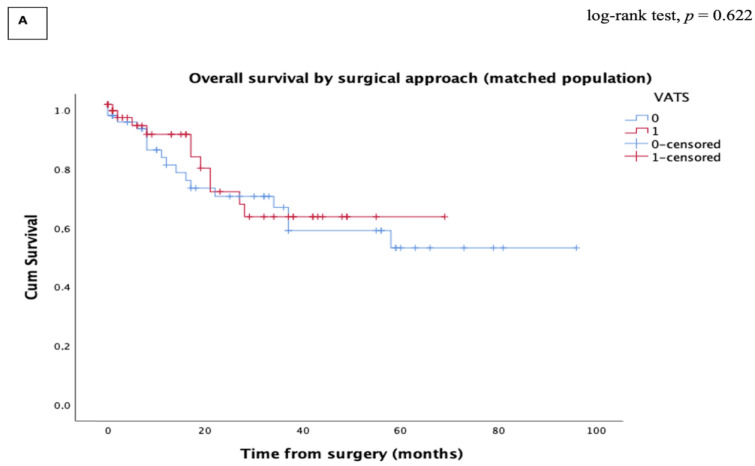
Kaplan–Meier curve for long-term survival after propensity score matching (PSM). (**A**) Overall survival; (**B**) Recurrence-free survival. VATS: Video-assisted thoracoscopic surgery.

**Table 1 cancers-15-00414-t001:** Patient characteristics for all included patients and PSM pairs.

Characteristics	All Included Patients			PSM Patients		
	Open	VATS	*p*-Value	Open	VATS	*p*-Value
Age, year ± SD	57.6 ± 9.52	61.33 ± 8.54	**0.031**	63.09 ± 9.9	62.03 ± 8.03	0.293
Sex, *n* (%)			0.664			0.695
Male	82 (64.1)	47 (61)		38 (59.3)	36 (56.2)	
FEV1%, mean ± SD	90.50 ± 16.64	88.94 ± 16.98	**0.008**	87.16 ± 12.8	88.35 ± 17.2	0.579
DLCO%, mean ± SD	65.92 ± 13.76	67.34 ± 15.24	0.599	61.7 ± 12.56	68.09 ± 16.43	0.105
Comorbidities, *n* (%)						
Hypertension	28 (21.9)	19 (24.7)	0.739	19 (29.6)	17(26.5)	0.653
Diabetes	17 (13.3)	7 (9.1)	**0.041**	7 (11)	7 (11)	1
Cardiac disease	21 (16.4)	5 (6.5)	**0.039**	5 (7.8)	3 (4.6)	0.358
COPD	25 (19.5)	19 (24.7)	0.385	10 (15.6)	12 (18.7)	0.633
Smoking history, *n* (%)	103 (80.5)	65 (84.4)	0.106	51 (79.7)	54 (84.4)	0.080
cTm size, mm ± SD	54.12 ± 35.77	31.16 ± 17.33	**0.002**	37.84 ± 20.8	35.01 ± 17.2	0.714
cTNM, *n* (%)			**0.033**			0.435
IIIA	77(60.2)	58 (75.3)		43 (67.2)	48 (75)	
IIIB	51 (39.8)	19 (24.7)		21(32.8)	16 (25)	
Induction therapy, *n* (%)	128 (100)	77 (100)	1	64 (100)	64 (100)	1
Time from preoperative therapy to surgery (days)	95 (72–135)	91(72–128)	0.723	92 (72–130)	89 (72–126)	0.642
Tumor location, *n* (%)			0.880			0.983
Right upper	50 (39.1)	32 (41.6)		27 (42.2)	25 (39.1)	
Right middle	4 (3.1)	4 (5.2)		3 (4.7)	4 (6.3)	
Right lower	21 (16.4)	15 (19.5)		11 (17.2)	10 (15.6)	
Left upper	34 (26.6)	17 (22.1)		15 (23.4)	17 (26.5)	
Left lower	16 (12.5)	10 (13)		8 (12.5)	8 (12.5)	
Surgical procedure						
Lobectomy	47 (36.7)	59 (76.6)		24 (37.5)	48 (75)	
Bilobectomy	28 (21.9)	10 (13)		13 (20.3)	8 (12.5)	
Pneumonectomy	32 (25)	3 (3.9)		16 (25)	3 (4.7)	
Sleeve lobectomy	21 (16.4)	5 (6.5)		11 (17.2)	5 (7.8)	

VATS: Video-assisted thoracoscopic surgery; SD: standard deviation; FEV1: forced expiratory volume in 1 s; DLCO: diffusing capacity of carbon monoxide; COPD: chronic obstructive pulmonary diseases. *p* values in bold are statistically significant.

**Table 2 cancers-15-00414-t002:** Treatment and tumor characteristics for all included patients and PSM pairs.

Characteristics	All Included Patients			PSM Patients		
	Open	VATS	*p*-Value	Open	VATS	*p*-Value
Induction therapy, *n* (%)						
Platinum doublet therapy only	90 (70.3)	48 (62.4)	0.282	39 (60.9)	38(59.4)	1
Platinum doublet therapy + Immunotherapy	5 (3.9)	3 (3.9)	1	5 (7.8)	3 (4.7)	0.717
Chemoradiotherapy	33 (25.8)	26 (33.7)	0.265	20 (31.3)	23 (35.9)	0.708
pTm size, mm ± SD	42.3 ± 30	31.4 ± 18.9	**0.002**	33.2 ± 23.1	30.2 ± 16.3	0.696
yp tumor stage, *n* (%)			**<0.001**			0.488
T0	14 (10.9)	11 (14.3)		8 (12.5)	10 (15.6)	
T1	23 (18)	33 (42.8)		18 (28.1)	25 (39.1)	
T2	34 (26.5)	16 (20.8)		16 (25)	14 (21.9)	
T3	37 (29)	13 (16.9)		15 (23.4)	12 (18.7)	
T4	20 (15.6)	4 (5.2)		7 (11)	3 (4.7)	
yp nodal stage, *n* (%)			**0.013**			0.227
N0	76 (59.3)	55 (71.4)		30 (46.9)	46 (71.8)	
N1	27 (21.1)	12 (15.6)		16 (25)	10 (15.6)	
N2	25 (19.5)	10 (12.9)		18 (28.1)	8 (12.5)	
Histology			0.910			0.983
Adenocarcinoma	76 (59.4)	48 (62.4)		38 (59.4)	37 (57.8)	
Squamous cell	47 (36.7)	26 (33.8)		24(37.5)	25 (39.1)	
Other	5 (3.9)	3 (3.8)		2 (3.1)	2 (3.1)	
Lymph nodes, mean ± SD						
Total stations	5 ± 1.9	5.5 ± 1.6	0.078	4.8 ± 1.6	5.9 ± 1.7	**0.011**
Total lymph nodes	10.7 ± 6.7	12.2 ± 5.1	0.400	11.5 ± 6.3	12.5 ± 5.3	0.219
Completeness of resection, *n* (%)						
R0	105 (82)	73 (94.8)	**0.012**	58 (90.6)	60 (93.7)	0.443

VATS: Video-assisted thoracoscopic surgery; SD: standard deviation. *p* values in bold are statistically significant.

**Table 3 cancers-15-00414-t003:** Perioperative outcomes of all included patients and PSM pairs.

Perioperative Event	All Included Patients			PSM Patients		
	Open	VATS	*p*-Value	Open	VATS	*p*-Value
Conversion to open, *n* (%)		12 (15.6)			10 (15.6)	
Any complication, *n* (%)	50 (39.1)	23 (29.9)	0.183	21 (32.8)	13 (20.3)	0.109
Air leak > 5 days	14 (11)	10 (12.9)	0.533	5 (7.8)	5 (7.8)	1
Atrial arrhythmia	6 (4.7)	3 (3.9)	0.879	3 (4.7)	2 (3.1)	1
Pneumonia	11 (8.6)	5 (6.5)	0.786	4 (6.2)	3 (4.7)	1
ARDS, *n* (%)	6 (4.7)	2 (2.6)	0.713	4 (6.2)	2 (3.1)	0.679
BPF, *n* (%)	6 (4.7)	0 (0)	0.148	3 (4.7)	0 (0)	0.244
Pleural effusion, *n* (%)	7 (5.4)	3 (3.9)	0.936	2 (3.1)	1 (1.6)	1
Reoperation, *n* (%)	12 (9.4)	3 (3.9)	0.129	3 (4.7)	1 (1.6)	0.619
Surgery time, min, median (range)	180 (60–520)	175 (60–430)	0.199	180 (60–348)	180 (60–330)	0.827
Blood loss, mL, median (range)	140 (50–2200)	130 (50–1200)	0.956	195 (50–1500)	140 (50–1100)	**0.023**
Chest tube duration, days, median (range)	3 (2–20)	2 (1–22)	0.311	3 (2–20)	2 (1–15)	0.160
Length of stay, days, median (range)	8 (3–40)	4 (2–45)	**<0.0001**	7 (5–40)	4 (2–35)	**<0.0001**
30 days in hospital death	2 (1.5)	1 (1.3)	1	1 (1.6)	0 (0)	1

VATS: Video-assisted thoracoscopic surgery; ARDS: Acute respiratory distress syndrome; BPF: Bronchopleural fistula. *p* values in bold are statistically significant.

**Table 4 cancers-15-00414-t004:** Patterns of recurrence of all included patients and PSM pairs.

Recurrence	All Included Patients			PSM Patients		
	Open	VATS		Open	VATS	*p*-Value
Overall *n* (%)	61 (47.6)	31 (40.3)	0.356	30 (46.9)	25 (39.1)	0.430
Local (mediastinum) *n* (%)	9 (7)	3 (4)		5 (7.8)	3 (4.7)	
Regional (lung) *n* (%)	16 (12.5)	5 (6.5)		7 (10.9)	3 (4.7)	
Distant *n* (%)	36 (28.1)	23 (29.8)		18 (28.2)	19 (29.7)	

VATS: Video-assisted thoracoscopic surgery.

**Table 5 cancers-15-00414-t005:** Multivariable Cox proportional hazards analyses for RFS.

Characteristics	All Included Patients			PSM Patients		
	HR	95% CI	*p*-Value	HR	95% CI	*p*-Value
Age	1.023	0.988–1.059	0.195	1.041	0.993–1.091	0.095
Sex (ref = female)	0.686	0.356–1.321	0.260	0.349	0.134–0.911	**0.031**
FEV1	0.998	0.979–1.017	0.815	0.980	0.953–1.008	0.158
DLCO	0.990	0.966–1.014	0.413	0.979	0.945–1.015	0.253
Comorbidities						
Hypertension	1.024	0.485–2.164	0.950	0.543	0.218–1.351	0.189
Diabetes	1.348	0.451–2.024	0.593	0.716	0.117–1.365	0.717
Cardiac disease	1.240	0.495–2.104	0.646	1.347	0.716–1.639	0.125
COPD	0.654	0.325–1.314	0.233	2.502	0.949–3.598	0.064
Smoking history	0.999	0.984–1.014	0.903	1.002	0.982–1.023	0.860
cTm size	0.985	0.968–1.001	0.074	0.950	0.921–0.979	**0.001**
Histology						
Adenocarcinoma (ref)	2.919	0.297–3.671	0.358	2.441	0.464–3.327	0.165
Squamous cell	1.381	0.494–2.608	0.167	2.126	0.385–3.554	0.199
VATS (ref)	1.589	0.812–2.109	0.176	2.359	0.946–2.883	0.066
pTNM	1.049	1.022–1.077	**<0.001**	1.092	1.050–1.136	**<0.001**

*p* values in bold are statistically significant.

## Data Availability

Data collected for this study will be made available by the corresponding author upon reasonable request following publication.

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
