# Peer review of "VATS versus Open Lobectomy following Induction Therapy for Stage III NSCLC: A Propensity Score-Matched Analysis"

_cancers, 2023, doi:10.3390/cancers15020414_

Round 1

Reviewer 1 Report

This is an interesting study comparing VATS and open lobectomy in stage III NSCLC. Interestingly, you included only patients with induction chemotherapy. As usual in this kind of study, you included different clinical scenarii (large T4 or N2, or central tumor). There is of course a big selection biais (most difficult cases realized by 1) experienced surgeon and 2) open apporach). :

- How did you choose the surgical approach, what were the criteria (surgeon experience, only single station N2?) please explain.

Methods section should be more precise for inclusion criteria.

- Did you inlcude bulky N2, large T4, bronchovascular invasion. How many patients had sleeve (bronchial or arterial) resection? this information should be mentionned.

- How many surgeons realized the VATS approach? What was the previous experience?

- I think you should change the title and include the term "neo-adjuvant treatment" instead of stage III. probably more relevant for readers.

ypT and ypN should be changed in the table 2 (instead of pT or pN, since all patients underwent neo-adjuvant treatment!)

Based on your experience and results, could you propose which patients are more suitable for VATS? please explain in the discussion

Author Response

Reviewer 1 : Thank you for your comments and requests

Comment 1 : This is an interesting study comparing VATS and open lobectomy in stage III NSCLC. Interestingly, you included only patients with induction chemotherapy. As usual in this kind of study, you included different clinical scenario (large T4 or N2, or central tumor). There is of course a big selection bias (most difficult cases realized by 1) experienced surgeon and 2) open apporach). :

- How did you choose the surgical approach, what were the criteria (surgeon experience, only single station N2?) please explain.

Methods section should be more precise for inclusion criteria.

Answer 1 :

The surgical approach was determined according to the size and the location of the tumor. Large central tumors with probable vascular invasion were offered thoracotomy. Peripheral tumors less than 7 cm were managed by VATS.  Since 2015, in case of chest wall invasion, a hybrid approach (VATS + an elective approach centred on the resected wall) was performed as described in reference 15.

Change 1 :  

L 83 : Patients with central tumors requiring complex bronchovascular reconstruction were offered an open approach. Peripheral tumors less than 7 cm were managed by VATS.  Since 2015, in case of chest wall invasion, a hybrid approach was performed as described previously.

Comment 2 :  Did you inlcude bulky N2, large T4, bronchovascular invasion. How many patients had sleeve (bronchial or arterial) resection? this information should be mentionned.

Answer 2 :

Patients with bulky N2 received surgery in case of a significant response to induction therapy and postinduction downstaging confirmed by EBUS or mediastinoscopy. Theses patients were managed by VATS. Five patients underwent sleeve lobectomy and 3 patients underwent pneumonectomy in the VATS group Versus 21 sleeve lobectomies and 32 pneumonectomies in the thoracotomy group in the entire cohort. More details are reported in table X    

Change 2 :

L89 : In patients with cN2 (single or bulky) and cN-3 disease, invasive restaging using mediastinoscopy or EBUS was performed after induction therapy and prior to resection. Patients with downstaging and without progression underwent radical resection.

Table X    

Comment 3 : How many surgeons realized the VATS approach? What was the previous experience?

Answer 3 :

In our institution the use of VATS for early-stage NSCLC started in the late 1990s (1). This experience prompted us to consider using VATS in complex post-chemotherapy, central tumors, chest wall and spine resections.  Four senior surgeon with at least 4 years VATS experience were in charge of the patients.

Change : 

L 92 : Four surgeons with at least 4 years VATS experience were involved in the study.

Comment 4 : I think you should change the title and include the term "neo-adjuvant treatment" instead of stage III. probably more relevant for readers.

Answer 4 :

Thank you for this helpful comment.  As you know, we are unfortunately restricted by the maximum number of words at submission time.

Change :

Title : VATS Versus Open Lobectomy Following Induction Therapy For Stage III NSCLC: A Propensity Score-Matched Analysis

Comment 5 :  ypT and ypN should be changed in the table 2 (instead of pT or pN, since all patients underwent neo-adjuvant treatment!)

Answer 5 : Thank you for pointing it out.

Change 5 : Table X

Comment 6: Based on your experience and results, could you propose which patients are more suitable for VATS? please explain in the discussion

Answer 6 :

Clearly, The indications of VATS depend on each surgeon experience (2,3). In our pratctice the only cases for which we do not offer VATS are : centraly located tumors with vascular invasion and tumors requiring pneumonectmoy.   We are strongly convinced that the development of robotic surgery and the accumulation of «  surgical robotic skills »  will result in a widening of minimally invasive techniques  for  large and central tumors in the coming years.

Change 6 :

L 272 : We believe that VATS approach should be reserved for tumors ≤7 cm and tumors without  massive hilar invasion .Chest wall involvement or N2 disease should not hamper the VATS procedure, even after induction therapy

1- Video-assisted thoracoscopic lobectomy: an unavoidable trend? A retrospective single-      institution series of 410 cases. Marty-Ané CH, Canaud L, Solovei L, Alric P, Berthet JP.Interact Cardiovasc Thorac Surg. 2013 Jul ; 17(1):36-43. doi: 10.1093/icvts/ivt146.

2- Technique and outcomes of 79 consecutive uniportal video-assisted sleeve lobectomies Soultanis KM, Chen Chao M, Chen J, Wu L, Yang C, Gonzalez-Rivas D, Abu Akar F, Jiang G, Jiang L.Eur J Cardiothorac Surg. 2019 Nov 1;56(5):876-882. doi: 10.1093/ejcts/ezz162.

3- D Xie J Deng D Gonzalez-Rivas 2020 Comparison of videoassisted thoracoscopic surgery with thoracotomy in bronchial sleeve lobectomy for centrally located non-small cell lung cancer J Thorac Cardiovasc Surg https://doi.org/10.1016/j.jtcvs.2020.01. 105

Reviewer 2 Report

Congratulations to the authors for the work which, in my opinion, offers interesting insights into the correct surgical approach to stage IIIA lung cancer. It is true that there are many limitations, but in any case I find it well structured and with a certain significance. The results clearly show how the VATS approach in expert hands has become a safe and effective method with significant benefits for the patient (QoL and pain) and for the hospital (days of hospitalization).

In this regard, if possible, I would like to ask for some further information:

1) How can so many extra days of hospitalization be justified in the Open technique if the days of drainage are similar (+/-1)?

2) The surgical choices are personal to the operator, but how many surgeons are involved in the study and with which VATS experience?

Finally, in the discussion and in the conclusions, I would like to underline how the VATS approach is indicated for lesions with a minor T and that patient selection is also fundamental for reducing conversions.

Thanks and good job

Author Response

Reviewer 2 : Thank you for reviewing the article

Comment 1 :   How can so many extra days of hospitalization be justified in the Open technique if the days of drainage are similar (+/-1)?

Answer 1 :

Indeed, the outcomes of hospitalization duration are surprising. One possible explanation is reduced trauma leading to less postoperative pain and  lighter psychological burden in the VATS group.

Change 1:

L 286 : One possible explanation is the reduced trauma leading to  less postoperative pain and  lighter psychological burden in the VATS group allowing  faster recovery  than open lobectomy.

Comment 2 : The surgical choices are personal to the operator, but how many surgeons are involved in the study and with which VATS experience?

Answer 2 :

In our institution, the use of VATS for early-stage NSCLC started in the late 1990s (1). This experience prompted us to consider using VATS in complex post-chemotherapy, central tumors, chest wall and spine resections. The surgical choices are not realy personal, all cases are collegially discussed by the surgical staff.  Four senior surgeon with at least 4 years VATS experience were in charge of the patients.

Change 2 :

L 92 : Four surgeons with at least 4 years VATS experience were involved in the study

Comment 3 : Finally, in the discussion and in the conclusions, I would like to underline how the VATS approach is indicated for lesions with a minor T and that patient selection is also fundamental for reducing conversions.

Answer 3 : Indeed, as of now there is clearly a patient selection process favoring a VATS approach for smaller and more peripheral tumors. The indications and contraindications for lung cancer treatment have changed overtime. As you know, initially only early stages were considered for VATS approach and several concerns regarding the use of VATS for advanced stage have been raised for decades.  With the widespread use of VTAS, the recent series repporting have shown encouraging results  of VATS resection for central tumors (2,3) , the development of robotic surgery and the acquisition of « surgical robotic skills », we should expect  a widening of the recommendations of minimally invasive techniques  for  large and central tumors in the coming years .

Change 3 :

L 272 :  We believe that VATS approach should be reserved for tumors  ≤7 cm and tumors without  massive hilar invasion .Chest wall involvement or N2 disease should not hamper the VATS procedure, even after induction therapy.

L 338: The current study has demonstrated that in experienced VATS centers, VATS lobectomy for the treatment of advanced stage III NSCLC is safe, reliable, and is associated with a shorter hospital stay and equivalent oncological outcomes compared with standard thoracotomy. We believe that in selected patients without huge and/or central tumors, VATS may be a sound alternative to thoracotomy, especially for patients weakened by muti-modality treatment. Patients selection process is fundamental to avoid untimely conversions.

1- Video-assisted thoracoscopic lobectomy: an unavoidable trend? A retrospective single-      institution series of 410 cases. Marty-Ané CH, Canaud L, Solovei L, Alric P, Berthet JP.Interact Cardiovasc Thorac Surg. 2013 Jul ; 17(1):36-43. doi: 10.1093/icvts/ivt146.

2- Technique and outcomes of 79 consecutive uniportal video-assisted sleeve lobectomies Soultanis KM, Chen Chao M, Chen J, Wu L, Yang C, Gonzalez-Rivas D, Abu Akar F, Jiang G, Jiang L.Eur J Cardiothorac Surg. 2019 Nov 1;56(5):876-882. doi: 10.1093/ejcts/ezz162.

3- D Xie J Deng D Gonzalez-Rivas 2020 Comparison of videoassisted thoracoscopic surgery with thoracotomy in bronchial sleeve lobectomy for centrally located non-small cell lung cancer J Thorac Cardiovasc Surg https://doi.org/10.1016/j.jtcvs.2020.01. 105